# Territorial Development and Cross-Border Cooperation: A Review of the Consequences of European INTERREG Policies on the Spanish–French Border (2007–2020)

**Javier Martín-Uceda** *  **and Joan Vicente Rufí** *

Department of Geography, Universitat de Girona, 17004 Girona, Spain
* Correspondence: javier.martin@udg.edu (J.M.-U.); joan.vicente@udg.edu (J.V.R.)

**Abstract:** Territorial cohesion policies are a priority for the European Union. For over thirty years, they have aimed not only to provide greater social and economic development across all European space, but also to contribute to balancing g internal social and economic inequalities. On the other hand, European institutions have adopted regional scale as the optimal to achieve this broad goal. Consequently, the ability of these policies to solve the problems faced by some of these regions has been one of the most widely researched areas in numerous scientific disciplines. This article aims to assess the impact, over a fifteen-year perspective, of cooperation funds focusing on a specific area, the cross-border, and, in particular, the border area separating Spain and France. Specifically, the analyses of data from operative programmes IV and V of the INTERREG-A projects produces contradictory results. While the aim of European institutions was to use the European Territorial Cooperation instrument to achieve a greater, better real impact of funds in cross-border areas, and to progress towards territorial cohesion, the results show that, conversely, they have largely contributed to reinforcing unequal development. In the analysed border, the dynamics are an increasing distance between the more and less developed areas in the direct border space, and a privilege of urban areas, even if they are far from the borderline. A relevant conclusion of the text is that these unexpected results are partly a consequence of the design of the European programmes.

**Keywords:** cross-border cooperation; European polices; INTERREG; political geography

## 1. Introduction

The aim of this article is to analyse the real effects of operative programmes IV and V of the INTERREG-A projects on territorial cohesion policies, and their link with development in cross-border territories. This is the result of the research that has been carried out since 2012 by the APTA (Territorial and Environmental Planning and Analysis) group, which belongs to the Geography Faculty of the University of Girona, with the collaboration of researchers from a number of Spanish, French and Polish universities. The group has analysed just under 700 projects.

The text has four sections. The first provides the theoretical framework used in the research; the second covers the methodology and data used, as well as a brief introduction to the area studied; the third includes the results and their analysis; the fourth and final part presents the conclusions.

## 2. Cohesion and Sustainable Territorial Development. A Twofold Community Challenges

European territorial policy, in its varying forms over the decades, has had a clear aim, to further the consolidation of a more cohesive territory among the various regions that facilitates social and economic development beyond the large European systems and urban centres, and that should improve the quality of the inhabitants' lives [1–5]. This is a complex aim, given the great difference in the realities and contexts of territories in the north and south, and east and west of the continent, and this has been exacerbated by the

successive expansion in the number of European Union member states. The concept of cohesion has, therefore, been a subject of debate, both political and academic, between experts in the subject [1,6–8].

Regional policy is one of the fundamental pieces in the construction of the European Union, and was covered in the founding treaties of the community's institutions [9]. It was from 1975 that common policy became more explicit, partly through the creation that year of the Regional Development Fund, and, more specifically, the creation of the Cohesion Fund in 1994. The latter collected the joint action needed to face major territorial challenges that put the effective cohesion of the common market, and cohesion between the regions and European citizens, at risk [10,11].

The great complexity and diversity of territorial realities meant that those instruments arising from community policies needed to be highly adaptable if they were to meet the aim of cohesion [12] and sustainable development. Furthermore, they had to be approached from a cross-disciplinary position [5,13], adding to the reality further challenges resulting from global dynamics, such as climate change, socio-ecological transition, the financial markets, large-scale migration and other socio-political changes closely linked to globalization and the expansion of the advanced capitalist model and its forms of territorialization [14–17].

Indeed, some of the dualities that existed at the very time the Union European was founded had, far from being resolved, become more serious through internal and external processes. Selective urban dynamism, and, at the same time, the 'emptying', in demographic, sociological and all related cultural terms, of rural areas are, perhaps, the best examples of this dual model of European integration [18,19]. Some territories have benefitted greatly from this, while others are still unable to mobilize their resources and potential towards achieving a model of sustainable socio-economic development.

Such results are close to the 'Matthew effect'. This reality has also become evident in recent years in the political and social debate in numerous European states across the entire continent. The cases of Spain and France are good examples. The appearance of mobilizations that have an impact on the social and political debate has highlighted the reality of depopulation and the lack of economic opportunities for a sizeable part of the territory. In Spain, movements have arisen in provinces such as Teruel or Soria that centre the debate on precisely these issues, referring to what is called España Vaciada (Emptied Spain) [20,21]. The case of France is more complex, as movements such as the Guilettes Jaunes (Yellow Jackets) [22] focus not only on rural areas. While the movement is broader than in Spain, channeling wider protests that highlight deficits in the social, political and economic systems, one of its pillars is the territorial aspect and its demands.

In most European cross-border areas, one can see the customary isolation [23,24] caused by the distance from capital cities, peripheralization and the barrier effect [25] of frontiers; all of these have often hindered development and caused low demographic dynamism; exceptions are to be found in those few places where it is precisely this state limit that has been a motor for dynamism. This could be deemed 'normal,' as the consolidation of the Common Market under the 1957 Rome Treaties, the changing nature of frontiers resulting from the treaty of Schengen of 1985 and its progressive implementation from 1995, has led to a change in the role frontiers play. No longer are they distant, isolated spaces, they have become, in theory at least, a key element in the European Union [26,27]. Throughout this process of integration and cohesion, cross-border areas have acquired a vital role that is both real and symbolic; this prominence has been renewed over recent decades, and has become a 'laboratory' of European policy [28] for the feasibility of the single market, and also for development and cohesion policy.

In short, the case of cross-border territories is singular. From the perspective of development and cohesion policy, they unite two negative territorial dynamics—geopolitical peripheries and a crisis of territoriality. To combat this, European institutions have dedicated numerous resources and specific programmes that have been in place long

enough for their effects to be assessed. As a starting hypothesis, we believe that these effects are not exactly those hoped for, and may even be the opposite.

## 3. Territorial Challenges (Cross-Border) in the European Political Context; Territorial Agendas

While territorial planning is a competence of individual EU member states, community policy may play an important role in its development. The Single European Act of 1986 specified economic and social cohesion as a goal, but it was not until the treaty of Lisbon of 2007 that a third dimension was added, that of "economic, social, and territorial cohesion". Even before this explicit mention, 1999 saw the passing of the first European Spatial Development Perspective (ESDP), which began to design and articulate a proposal for European territory. It included ideas of authors such as Brunet [29] regarding a territory that, as well as containing great differences, also contained great inequalities in its development dynamics and all social and economic dimensions. While it included some highly dynamic regions and cities, which formed part of international networks, it also included other regions in profound retrogression and/or isolation. Furthermore, and perhaps most relevant, these different and unequal ideas of territorial structuring transcended state borders and, thus, formed supra-state macro-regions [30–34]. Conversely, another central facet of the ESDP is the support for a poly-centrism [35,36] that foments development and cohesion, counterbalancing structures that are often very centralized and taking advantage of a historical legacy comprising urban centres spread far and wide, including the large rural areas of much of Europe.

Another important aspect of the ESDP is the specific recognition of border areas [37]. These have a dual reality. Most are peripheral not only to the dynamic centre of Europe, but also to the centres of each state; at the same time, there is the still a patent complexity in being able to connect with the "other side" of the border. Although there is no administrative reason for these borders to exist, they still mark the functional and psychological reality of relations in many aspects, such as the lack of infrastructures; a mutual lack of knowledge on many levels; a multi-level asymmetry [38] that makes them a wall in some cases and causes absolute dependency in others. For these reasons, along with their reality and their real and symbolic importance in the success or failure of the European project, cross-border spaces and cooperation would benefit after 1989 from their own economic instrument, called INTERREG that, as will be shown below, deserves a special mention in the policies of cohesion.

The ESDP thus set a precedent regarding various territorial problems; this would be extended and detailed in the two following European territorial agendas. Two of these problems are of particular interest in the case of this article. The first is the governance and administration of cross-border areas; the second is sustainable territorial development and, in particular, its social and economic component.

The first of these agendas, "The Territorial Agenda of the European Union Towards a More Competitive and Sustainable Europe of Diverse Regions," was agreed in the German city of Leipzig in 2007 [39]. The title makes explicit reference to some of its priorities, competitiveness and sustainability, and acknowledges the reality of regional diversity. The document sets out six territorial challenges, some of which resulted from the Union's extension eastwards in 2005. Of the six challenges, the following three stand out:

- The impact of climate change on sustainable development;
- The unequal integration of regions, including cross-border ones;
- Territorial effects of demographic change, in particular aging.

In line with the Agenda's title, the document develops six priorities to tackle the above-mentioned challenges. The strengthening of governance between stakeholders is worth special attention in that it aims to foment polycentric development, to establish a new relation between urban and rural areas and for the sharing of the administration of natural, patrimonial and landscape risks and resources.

In the same context, the "Territorial Agenda of the European Union 2020" [40], passed in Hungary in 2011, is subtitled "Towards an Inclusive, Smart and Sustainable Europe of Diverse Regions". As can be seen, reference is made to two of the elements present in the 2007 Agenda, namely sustainability and diverse regions. This document renews the earlier Territorial Agenda, while territorialising some of the key aspects of the EU 2020 Strategy (EU, 2010) and the treaty of Lisbon (2007).

A first aspect of the document to highlight is the insistence on the concept of territorial cohesion. Point 8 defines this as "a set of principles for harmonious, balanced, efficient, sustainable territorial development. It enables equal opportunities for citizens and enterprises, wherever they are located, to make the most of their territorial potentials" (2011, p.3). As a condition for this development to be sustainable and widespread, emphasis is placed on the need to promote development across all regions, adapting to the reality of each territory and fostering its local characteristics. Consequently, on setting out its priorities, it refers to the need to foment sustainable development while bearing in mind the unique characteristics of an individual territory, whether rural or cross-border. It also mentions the need to promote local capitals in order to guarantee long-term competitiveness and solutions. It provides support to the specific case of cross-border areas by saying that "European Territorial Cooperation should be better embedded within national, regional and local development strategies" (2011, p. 7).

The agendas ultimately characterize cross-border regions and relations as a particularly problematic issue case within a European territory that is, in itself, complex. These handicaps include the lack of cohesion and sustainable development, the context of climate change and huge environmental challenges, problems of demographic regression and the weakness implicit in articulating an indispensable formula of governance. As has been said, this diagnosis implies the articulation of specific policies and instruments for the cross-border regions that this article studies.

## 4. The Paradigm Shift of European Territorial Cooperation: New Instruments and Priorities for Cross-Border Cooperation

Continuing with the desire to pursue the reduction in inequalities between regions, and, in particular, the elimination of the barrier effect between internal borders, 1989 saw the creation of the INTERREG fund [41]. This is financed by European Regional Development Funds (ERDF), and its legal status has changed over the various programmes [42]. Each INTERREG programme lasts as long as the economic programmes established by the European Union, normally six years. The funding assigned to INTERREG has increased with each programme, at the same time as the number of European Union frontiers has increased with the adhesion of new member states.

The INTERREG fund is divided into three funding lines [43]. The first is for cross-border cooperation (territories either side of a border); the second is for trans-national cooperation (large regions or macro-regions); the third is for inter-regional cooperation (the creation of cooperation networks). Since the fund is promoted by the European Commission, it is EU institutions rather than member states that have the decision-making powers, even if the latter are the administrating authority [44]. The funds are administered through operative programmes [45] which are structured by the subsidized territories themselves. This study will analyse the first line of funding, that of cross-border cooperation.

The success of the first INTERREG programmes was hindered by challenges of cross-border governance; this led the European Parliament to create the European Grouping of Territorial Cooperation (EGTC) in 2006 [46]. This was a new instrument designed to foment cooperation, improving a number of aspects that had become an obstacle to cooperation, such as legal and administrative asymmetry between states. It permits all agents, of whatever territorial sizes, to create structures of cooperation, facilitating their institutionalization [47,48].

Lastly, in order to gain visibility and weight among the array of European funds, major changes were introduced regarding cross-border cooperation and the INTERREG fund in the 2014–2020 funding period. Specific regulations were adopted, and the European

Territorial Cooperation (ECT) became an aim of the ERDF, one of the three pillars of the policy of cohesion, alongside convergence and competitiveness, gaining visibility in the morass of European financial instruments.

In summary, during the first decade of the 21st century, just before the 2008 financial recession, territorial policy in Europe was sufficiently well-defined and seasoned to dispose of large (and growing) resources, ad hoc instruments and clear aims focused on territorial cohesion, sustainable development and climate change. If the cross-border factor is added to this policy, the aims do not change, in fact the opposite is true. Meeting these aims in cross-border areas would perhaps be the greatest symbol of integration and cohesion. Overcoming the physical, functional and mental barriers, the asymmetries they generate, and their peripherality, would show the efforts of the European project to be an extraordinary success. This research arises from such a position, analysing almost 15 years of interventions in contexts which appear to have worsened rather than improved. What has the contribution of the INTERREG programmes been, and what have they contributed to?

## 5. Methodology and Data Used

The methodology used in this article is sound, given that it has been tested and approved by two research teams led by researchers from the Territorial and Environmental Planning and Analysis group of the Geography Faculty of the University of Girona. More specifically, the research undertaken has analysed the INTERREG-A projects 2007–2013 for the borders of Spain and France, Spain and Portugal, Poland and Germany, Italy and Austria and Poland and the Czech Republic; for the Spanish–French border in projects covering the 2014–2020 period, and currently for the Spanish-Portuguese border. To give some idea of the breadth of this research, 688 projects have been analysed and investigated.

### 5.1. Methodology and Analysis Undertaken of European Borders

The analysis was carried out using data provided by the Programa Operativo España-Francia-Andorra (POCTEFA) (Spain-France-Andorra (POCTEFA 2014–2020) cooperation programme) [49], which included the name of the Project and its stakeholders, as well as information regarding who led it and was therefore responsible for its management and administration. Each Project was given a category depending on a typology that had been established beforehand by the research group. Nine types were created: local economic development; environment; research; culture and education; accessibility and transport; territorial planning; health, social cohesion and integration; security.

Each of the stakeholders is given a geographical code below, this is a fundamental step as one of the project's aims was to provide a cartographic dimension to the cooperation and its results. The codes coincide with NUTS European nomenclature, classified as NUTS II, III and LAU2. In Spain, this corresponds to the autonomous community, province and municipality, respectively. In some cases, new codes were needed in order to map stakeholders that did not fit in these territorial scales, such as counties, groups of municipalities or Euro-regions. Areal representation was chosen for the cartographic rendering of projects and stakeholders, which entails the use of layers of polygons that correspond to the area of each territorial stakeholder. Some stakeholders were more difficult to classify, such as the universities, research centres or chambers of commerce, as they may have varying degrees of territorial influence. Ultimately, it was decided to give them a NUTS III classification.

Giving each stakeholder a code and geographical ambit enabled the simultaneous representation of several stakeholders, on several territorial scales, although it did entail some map algebra calculations beforehand. One of the methodological problems that was posed was the complexity of representing, on the same information level, one or more stakeholders whose areas partly coincided. An example is when representing the stakeholders in a project involving the municipality of Girona and the province of Girona. The solution was map algebra, basically the sum of the various levels representing the stakeholders. Thus, the sum of the municipality of Girona (value = 1) and the province of Girona (value = 1) took into account both stakeholders and their territorial area, and

reflected the double participation of the ambit of the municipality of Girona (value = 1 + 1 = 2), given that it is already included in the province of Girona.

The final result, together with the cartography, produced a collection of data bases, with the different aspects analysed for the projects and their types, the agents and the territorial codes.

*5.2. The Spanish–French Border: A Brief Profile*

The Spanish–French border was fixed under the Treaty of the Pyrenees of 1659 [50,51]. With a length of 656km, it can be divided into two parts. There is the long, more mountainous central section, with little demographic dynamism, a small rural population that has moved from agricultural economies (and the occasional industrial enclave) to others more focused on the economic development offered by tourism. Then, there are the Atlantic and Mediterranean extremities, sites of the main natural, historical and infrastructure border crossings. The main urban centres are found here, with a more dynamic economy, particularly on the Atlantic side. No profile of the border can omit the state of Andorra, with an area of 468 km$^2$ and a population of almost 80,000. While, for this article, Andorra is not studied, from an industrial, economic and functional perspective, it plays an important role in cross-border activity, particularly so in the case of the border area with Spain [52].

Administrative differences exist on the two sides of the border, this is particularly so in the case of the competences of authorities, whether regional (NUTS II), provincial or departmental (NUTS III) or local. This would play a role in the administration of the INTERREG projects, as shown below. On the NUTS II level in Spain are the Basque Country, Navarra, Aragon and Catalonia; all are regions with great political power, clear decentralization and, in at least three cases, a deep-rooted identity. Before the administrative reforms of 2015, on the same level in France were Aquitaine, Midi-Pyrénées and Languedoc-Roussillon. In 2016, the regions of Midi-Pyrénées and Languedoc-Roussillon merged to form the Occitanie region. Given the centralized nature of the French state, French regions enjoy less political and administrative power than those in Spain. The provinces and departments are the authorities that represent the central state in Spain and France, respectively, and play a supporting role at the local level; in France the departments take on some elements that the state has decentralized.

On a local level, there is a rural element in the majority of border-area municipalities studied [53,54]. The lack of urban centres of any real size is a notable reality; the only area with a certain urban density is the Basque Eurocity on the Atlantic coast. The big cities, such as Barcelona, Toulouse and Zaragoza are far from the border, while towns such as Perpignan, Pau and Figueres are a few dozen kilometers away. Many of the areas with the highest urban density are found around the road and rail communication links. The rest of the territory, determined by the Pyrenees, has a complicated connectivity between north and south, and this factor should be kept in mind regarding the finality and feasibility of cross-border projects. One could thus state that most of the border can be labelled "empty territories" (as opposed to "emptied", since very few areas have historically had a population or density of any real importance).

## 6. Results and Analysis

Having briefly explained the space studied and methodology used, we can now turn to the results of this research.

The first point to mention (Table 1) is the number of projects (327) and stakeholders (1731) analysed in both programmes (2007–2013 and 2014-2020). While there was a slight increase in the number of projects in the second programme, the number of stakeholders more than doubled.

**Table 1.** Investment, projects and stakeholders in POCTEFA IV and V.

| | 2007–2013 (POCTEFA-IV) | 2014–2020 (POCTEFA-V) | Total |
|---|---|---|---|
| Milions (M.€) | 168 | 177 | 345 |
| Projects | 154 | 173 | 327 |
| Stakeholders | 641 | 1090 | 1731 |

Source: POCTEFA. Authors.

Therefore, a first aspect of note is the increase in cross-border projects and the high number of participants.

Regarding the investment made, the figures reflect what, in our opinion, is the relatively modest scope and dynamics of the programmes. European investment in programme IV was 168 million Euros, and this figure rose slightly to 177 million in the following programme. European investment accounts for 65% of the investment in each project, the remainder being provided obligatorily by each stakeholder; therefore, the total investment was substantially higher than the above figures.

*6.1. Projects Types*

The second aspect analysed was their subject matter, defined according to the above-mentioned framework. As has been said, the projects respond to the priorities and needs determined by each operative programme; these, in turn, were established by the European Commission and the various community-level policy agendas. Their classification has followed the categories detailed in the Methodology section.

An analysis of the data shows some notable realities and significant changes over the two periods. The first note-worthy point, as can be seen in Figure 1, is that the "Local economic development" field is the main category in both periods, although the second saw a fall of 15% due to the greater diversification of fields. The use of these funds to promote economic activity, particularly that linked to tourism and the backing given to small and medium-sized companies, was a priority over the whole period. An important change stands out upon analysing the second category, "Environment". It occupied the second place in the first period but was equaled by "Research" in the second. Much of the subject matter of both categories is shared, since environmental management and climate change are priorities in both cases. Also worth mention is the increase in the number of projects covering "Social cohesion and integration".

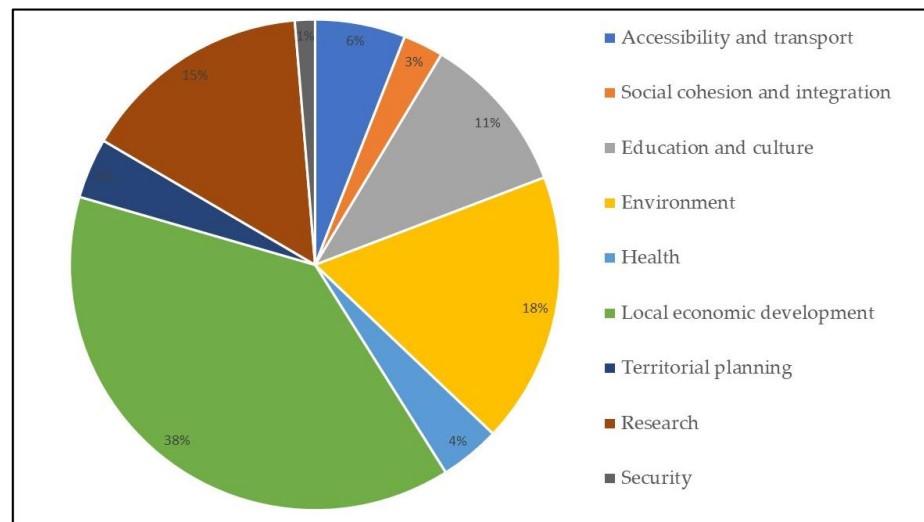

**Figure 1.** Graph. Project types. POCTEFA IV. Source: POCTEFA. Authors.

The remaining categories all account for a relatively low percentage, although this does not mean that they are less important or that they have lower impact or ability to raise funds. A pertinent example of this is the "Health" category, while a low percentage of projects cover this category, it has a high fund-raising ability. This can be seen in a more detailed analysis of data from the 2007–2014 period (Figure 2), which saw the building of the Cross-border Hospital of the Cerdanya, the largest investment made in the whole programme [43,49].

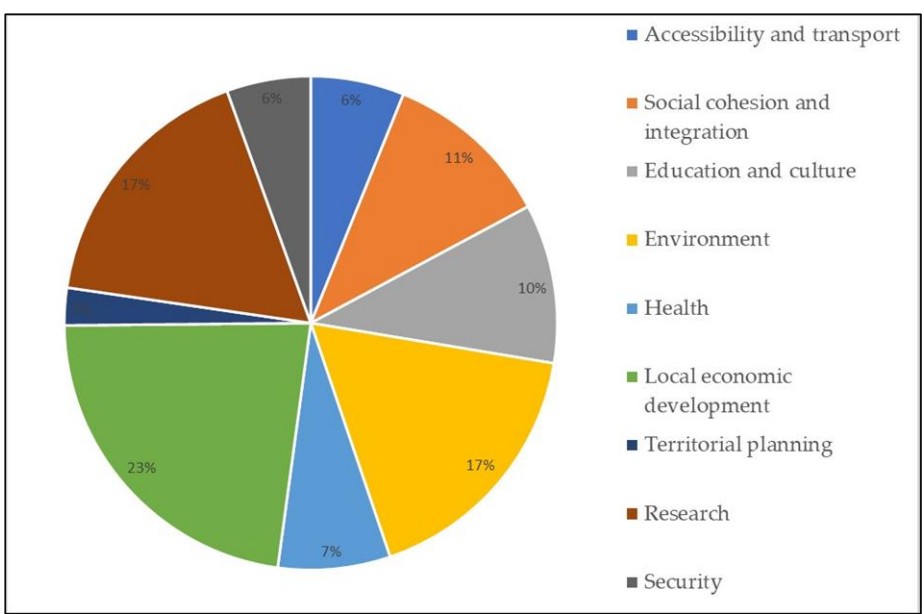

**Figure 2.** Graph. Project types. POCTEFA V. Source: POCTEFA. Authors.

The changes reflect a reorientation in the political priorities of the operative programme in line with the political horizon determined by the European Union. The rise in the number of research projects shows, a priori, the interest in making science and technology a cross-disciplinary aspect in all projects. This interest is more the result of the sudden arrival of universities as major stakeholders than any particular change in the priorities of local institutions. Indeed, the increased attention given to science and technology may well mean that such institutions find it harder to take part in the projects. This may also be true in the case of the more rural communities, by far the largest kind on this border, with greater limits to competition with towns and cities or other agents specifically dedicated to science and technology, such as universities.

### 6.2. Project Leaders

A further element analysed is that of project leaders. As was shown in the Methodology section, cartography permits the localization of territorial stakeholders and agents depending on their participation. An analysis of both maps of project leaders shows some shared contexts and some differences over both periods.

The first of the repeated aspects (Figure 3) is the greater capacity of Spanish in comparison to French stakeholders to lead projects; 70% of the leaders in POCTEFA IV were Spanish, while, in POCTEFA V, this figure fell to just over 61%. Despite this fall of almost 10%, the leadership capacity of Spanish stakeholders was still considerable, and clearly superior. Although the balance in the last project was slightly more even, over half of the leaders were still Spanish. The determining factor seems to be the greater political decentralization enjoyed by Spanish territorial agents. Institutional asymmetry, arising from one state being decentralized, and the other centralized, gives Spanish sub-state authorities greater political and administrative power. Other research [55,56] in other border contexts corroborates this thesis. Territorial agents in decentralized states lead more frequently than

those in centralized states. A good example of this is the participation, whether as leader or not, at the regional level. The Spanish autonomous regions participate widely in the projects, while the presence of French regions, both before and after the 2014 reform, is less. Institutional asymmetry, as will be seen throughout the analysis, is a key element in understanding a good part of cross-border dynamics.

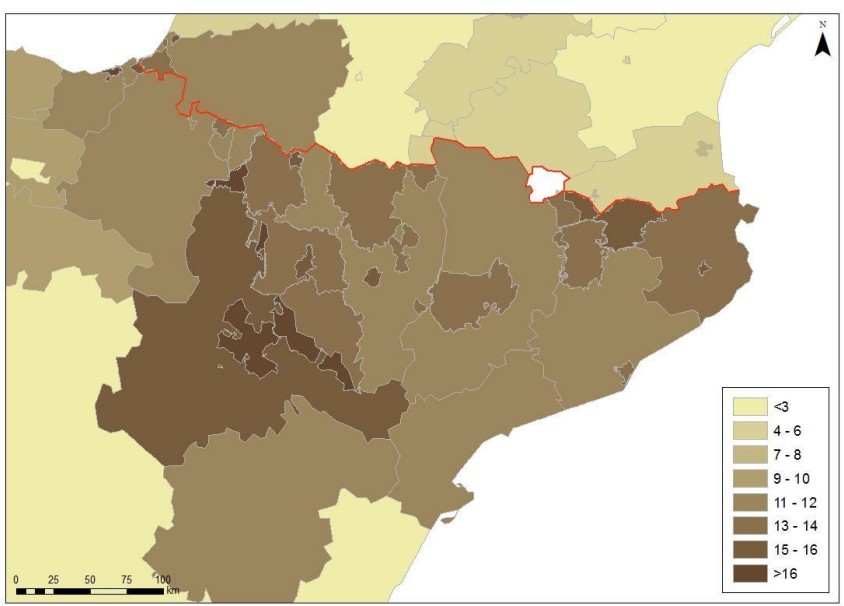

**Figure 3.** Map. Project leaders. POCTEFA IV. Source: POCTEFA. Authors.

The differences are even more evident in the participation at different territorial levels, whether local, provincial/departmental or regional. While, in programme IV, the local level stood out most, with important mobilizations in cities such as Zaragoza, San Sebastián and some counties in the north of Aragon and Catalonia, in programme V (Figure 4), a majority of projects were led at the departmental/regional level. This was particularly so in France, where 36 of 56 leaderships were at NUTS III, the departmental level. This fact is confirmed when analysing the participating agents, whether leaders or not. In Spain, it was Navarra, a single-province autonomous community that gave the regions (NUTS II) the leadership. If one counts its participation as a province, the NUTS III level would have been the most involved.

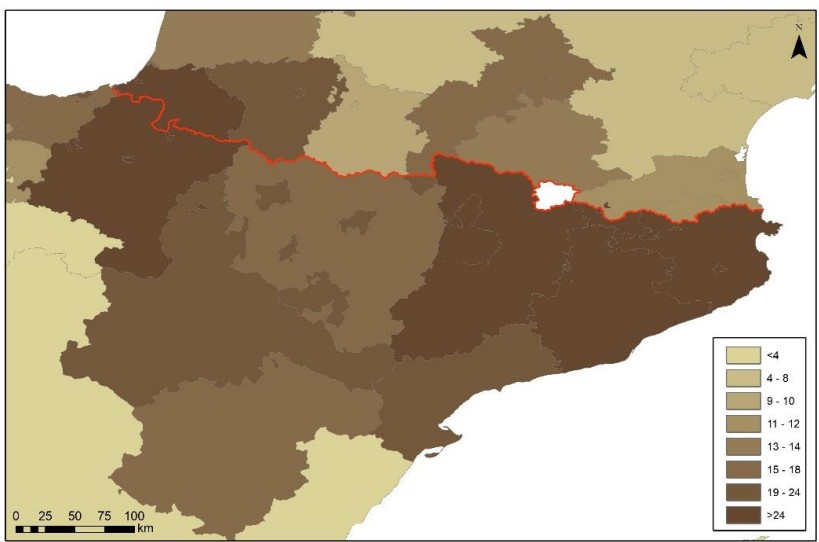

**Figure 4.** Map. Project leaders. POCTEFA V. Source: POCTEFA. Authors.

The change of level in leadership can also be seen in the typology of territorial stakeholder or agent. If most requests in programme IV were from municipal councils, programme V saw research parks and universities stand out. We have given these NUTS III level. The leadership of the universities of Zaragoza, Pau and Toulouse, alongside the technological parks in the Basque Country, is noteworthy. The highest funding was given to axis 1 of the programme, fomenting innovation and research, with practically 30% of all funding, as mentioned in the previous section. Research comes to the fore as one of the priorities when investing in projects.

A comparison of the two states shows that, in the case of France, it was departmental agents (NUTS III) that led most often over both programmes. This is due to the important participation of the universities of Pau and Toulouse. There is a notable change in Spain. In 2007–2014, local agents predominated, in particular the municipalities and counties of Aragon and Catalonia, and the municipalities of the Basque Eurocity, led by San Sebastián. However, in the 2015–2020 period, the provincial level assumed the role of leadership. This is due to an increased role of universities and research centres as agents promoting INTERREG projects.

### 6.3. Stakeholder Participation

Another aspect analysed was the group of stakeholders that participate, whether leading projects or not. The analysis showed an important change over the two periods. There were more French than Spanish agents in the 2007–2014 programme (Figure 5), while the reverse was true in the 2014–2020 programme (Figure 6). This latter programme provides evidence of the greater capacity of the Spanish to lead and take part in projects. One finding is repeated, NUTS III agents participate more in both programmes than other territorial levels.

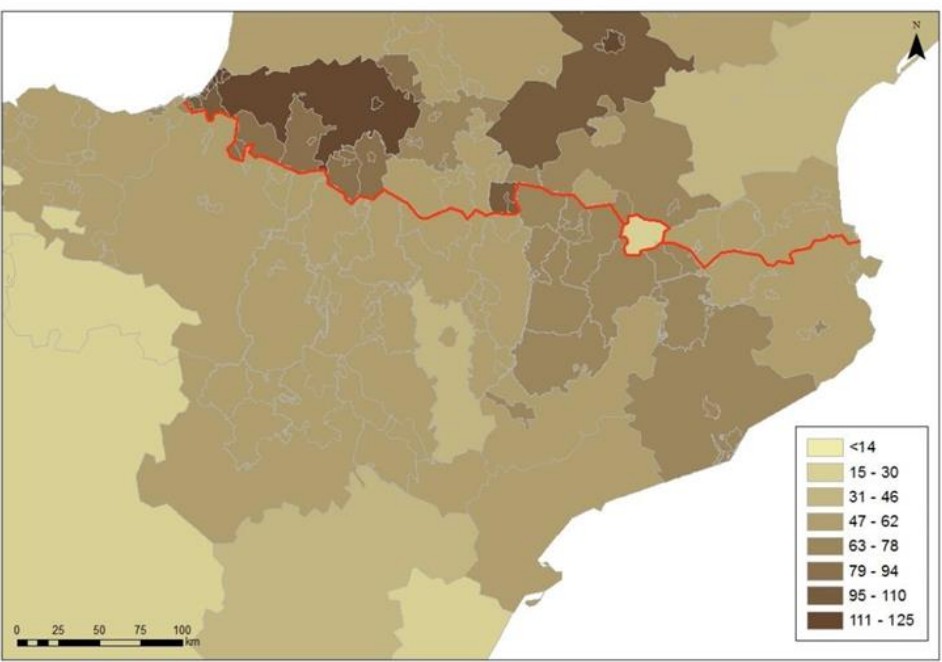

**Figure 5.** Map. Stakeholder participation. POCTEFA IV. Source: POCTEFA. Authors.

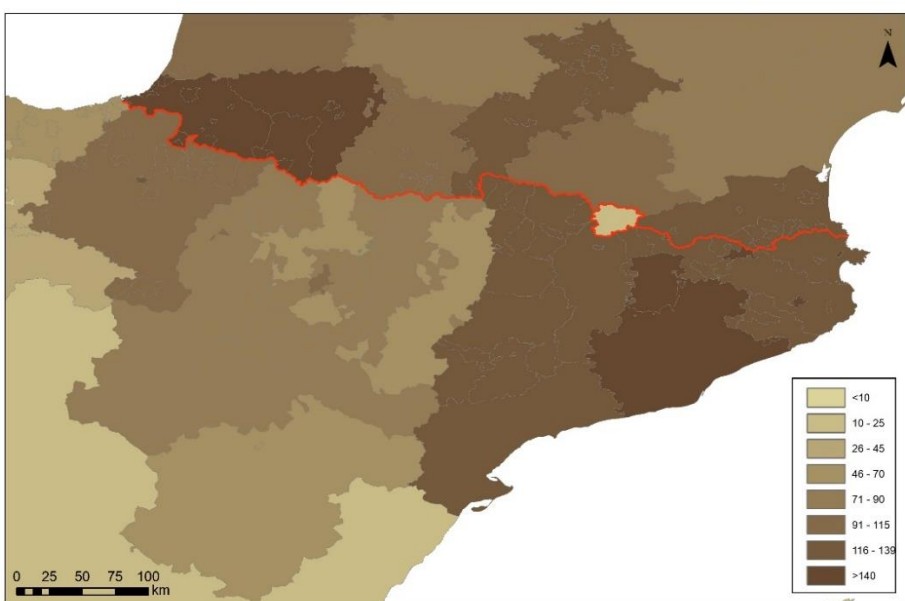

**Figure 6.** Map. Stakeholder participation. POCTEFA V. Source: POCTEFA. Authors.

On the French side of the border, the centre and the Atlantic coast host a majority of projects in both periods. The Pyrénées-Atlantiques departmental council, and the universities of Pau and Toulouse, have a highly developed capacity for participation in projects. The participation at the regional, NUTS II level, was minimal in the 2014–2020 programme, with 35 participations. However, there were 197 at NUTS III level, as opposed to 135 at the local level (LAU). This confirms the increasingly highly important role of universities and research centres as agents in cooperation.

The Spanish side of the border shows some features that distinguish it from the French side. Zaragoza and Barcelona stood out at the provincial level, NUTS III, in the 2007–2013 programme, also due to the high participation of their universities in projects. However, it was the regional, not provincial, stakeholders who took part in most projects. Navarra and Catalonia can be singled out, with public entities of their regional governments being involved in a range of projects in different fields. However, the balance in Spain was far more even than in France for the 2014–2020 programme. NUTS III was also the principal level, with 177 participations, as against 153 at the local level (LAU), and 148 at the regional level (NUTS II). Once more, universities play an important role in these statistics, particularly those located in Barcelona, Navarra and Zaragoza.

Another aspect worthy of analysis is the participation of local agents. At this level, due to its proximity to the population, the impact on the territory can be much greater. Furthermore, the link between European policy and local agents is an aspect that community authorities have promoted over recent years. The data in the following maps (Figures 7 and 8) show some changes between the two programmes. There was a wider variety of local stakeholders in period V, and this is particularly clear in those municipalities closest to the border, with greater participation on the Mediterranean side of the Pyrenees, in Navarra and the province of Huesca.

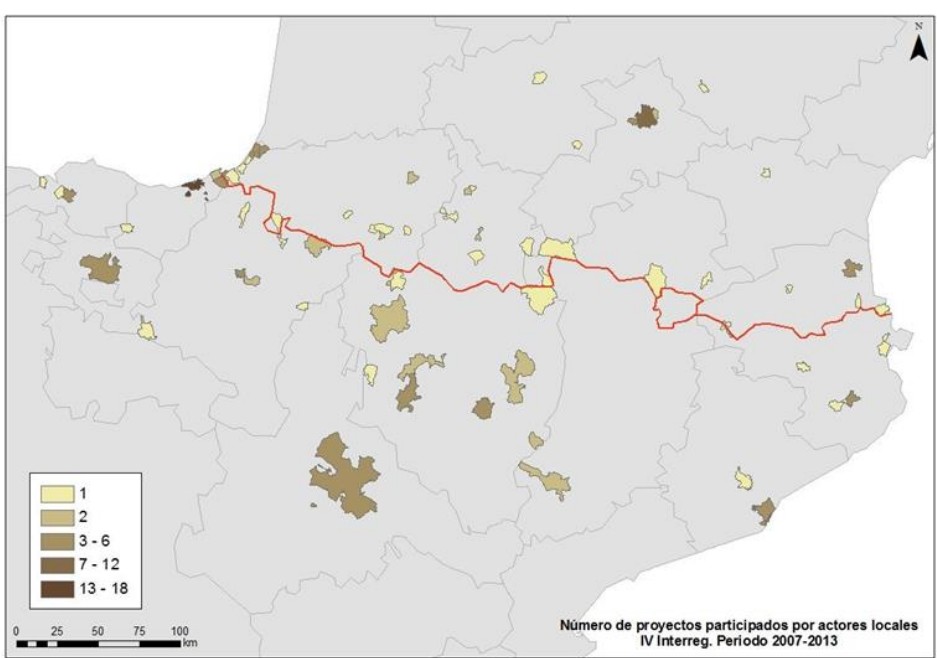

**Figure 7.** Map. Local stakeholder participation. POCTEFA IV. Source: POCTEFA. Authors.

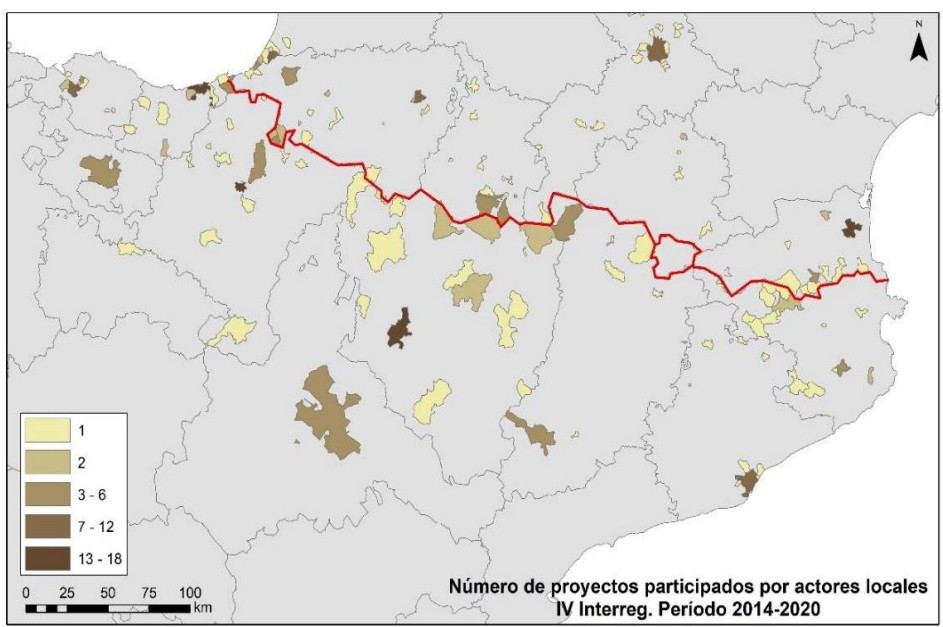

**Figure 8.** Map. Local stakeholder participation. POCTEFA V. Source: POCTEFA. Authors.

An analysis of the similarities reveals a constant, namely, the greater participation of the big cities of Barcelona, Zaragoza and Toulouse. These cities, over 100km from the border, are important stakeholders in cross-border cooperation. A further similarity is the Basque urban cross-border area continuing to operate as an axis of cooperation and projects; this is not seen with the Catalan urban axis, where variations are not found between programmes.

Regarding the group of stakeholders, whether project leaders or not, we see that Spanish agents became the majority during the period 2015–2020. However, this was not true of the previous call, where it was the French agents that participated more overall. The pattern of geographical distribution of stakeholders is the same in both periods. In France, most participations took place on the Atlantic coast and in the central border area,

around Toulouse. In the case of the southern side of the border, most participations were found in the Eastern part, in particular in the province of Barcelona. Once more, research centres and universities are highly important, and their participation increased greatly. The concentration of universities in large cities such as Barcelona or Toulouse supports this finding.

Regarding local stakeholders and agents, the greater participation of municipalities close to the border shows how projects can also be developed on a small scale, even if the networks are wide and have numerous stakeholders. Good examples of this in Spain are the Basque Eurocity, and the counties of the Ripollés area in Spain, and Vallespir in France. Although support for a cross-border territorial strategy would seem to be established here, a comparison of the two periods highlights the fact that large cities remain magnets for cross-border funding. As mentioned in the Data section, they are cities far from the border. This should cause us to reflect on functional border spaces and how far the "frontier effect" extends. Cooperation programmes should incorporate this perspective in order to make a real impact on the border territory.

## 7. Discussion and Conclusions

The results obtained from the data of the Spanish–French border show that the analysis of the real impacts of cross-border cooperation policies (and not only these) require a multi-scalar perspective, as Kaucic and Sohn recently pointed out [57]. It is essential to correctly measure these impacts to take into account sub-regional dimensions that give rise to differential effects in rural-urban areas, like a "truly" border-"technical" border realities, so that the very border-functional areas are clear as well as those that are only in administrative effects (as may happen with the regional scale). Observing other realities such as the area called PAMINA, in which Terlouw [58] reaches a very similar conclusion regarding where the real impacts of the policies are, leads us to consider that the case analyzed is not an exception, but perhaps a rule. The geographical and functional reality of PAMINA is very different from that of the Pyrenean border—without a mountain range that separates the different regions or with a significant urban density, indicating that there is a structural factor in these results which transcends in part characteristics-specific geographical areas.

This perspective can be analyzed in other European territorial realities. The difference between functional and administrative cross-border areas can alter the specific objectives of European policies and, at the same time, their instruments and models of implementation. Once again, we are facing a construction of European territory that requires a bottom-up perspective, which compensates for the predominant top-down dynamics.

The construction of the European territory in terms of development and cohesion implies a challenge for different European policies. The imbalances in place at the birth of the European project may have become less profound and changed regarding localization, motivation and content; however, they are still one of the defining characteristics of the project, despite the many strategies and actions undertaken over the decades that aim to harmonize a development that tends to be concentrated in a few, specific areas. These actions have become more defined and adapted themselves to new realities; despite this, they still find it difficult to have an impact on historical territorial inertia that favour the central areas of the continent, urban regions, centres of power or infrastructure corridors.

Among all of these issues, most border territories, generally uninhabited, still suffer the impact of secular isolation. Many of them have experienced, and still do, the paradox of being areas of symbolic transcendence; however, in practice, they are remote and peripheral in their own states. An exception to this is when geo-economic factors have made them points of contact and strategical exchange. These idiosyncrasies have been wrestled with in INTERREG programmes for the past 30 years; however, the challenges are still enormous, and institutional support is still needed. This support has mostly been economic; however, the past decade has seen the fomenting of mechanisms to facilitate administrative cooperation and to try to resolve administrative discontinuity.

As this article tries to show, asymmetries, in all their facets, but in particular those of a political nature, are sometimes restraints that are too strong to enable real harmonization of the border territory. In the case of the Spanish–French border, it is an imbalance that determines the participation of some agents and territories. The two periods studied confirm this fact. This leads to necessary reflection about how to structure mechanisms that provide facilities to all territorial agents, regardless of the organizational model of the state in question.

To a large extent, cross-border projects are the result of the parameters of the operative programmes. While it is true that these attempt to respond to territorial needs, they are also the reflection of the horizons established by community administrations. The approach to the prioritization and approval of projects is top-down. This leads to a degree of harmonization in Europe as a whole, but also to a greater difficulty in responding to the real needs of specific territories. In the case analysed, this has resulted in a distance between the majority of the territory and some of the subjects included in the operative programmes. The greater orientation, for example, towards scientific and technological projects, distances the rural world from the possibility of using European cooperation funds for their particular needs. In addition, the orientation of operative programmes towards certain aspects of development has led to the emergence of competition between territorial agents and stakeholders. In the case analysed, the bigger cities, while far from the border, have greater capacity to participate in the projects, exacerbating territorial duality and partly increasing the isolation of part of the territory. The projects, thus, fail to make a decisive contribution to the sustainability of border territory. This focus on top-down project administration does not facilitate the promotion of integrated and sustainable development strategies.

Giving a different perspective to European funds, that is closer to the territory, would perhaps result in them having a greater real impact in border areas. By this, we mean that it would be necessary to find formulas that motivated and permitted bottom-up projects. A vital condition to such projects would be to provide the more local entities and stakeholders with a greater capacity for reflection and preparation of projects that could compete with institutions on other territorial scales, and with far greater technical capacities and political power.

Another aspect resulting from this vision is that of functional areas or zones that are eligible for cross-border funding. The European Union's current limit of the border area includes those NUTS III adjacent to the border, and, in some cases the participation of other NUTS III close to the border is permitted. The use of even adjacent NUTS III may be a boundary that does not coincide with the functional areas, those that are most affected by the frontier effect. The larger the subsidizable area, the greater the completion between agents, and the lower the direct impact of funds and projects on the frontier territory. In other words, the recognition and stimulus of supra-municipal but sub-departmental/provincial territorial scales may make the real impact of projects on border territory more effective. If, on the one hand, the creation of EGCTs was an important step forward in administrative terms, the recognition of supra-municipal entities and initiatives would be a quantum leap. There are signs that some of these concepts will be incorporated in the new regulations on European funds for funding period 2020–2027; however, a closer examination is necessary. All of this should mean that European cross-border policies recognize that not all border areas are equal, and that the strict and homogenous application of available instruments is therefore neither possible nor efficient.

In summary, it seems unavoidable to continue to adapt cross-border development and cohesion policies given the correct analysis of the results obtained so far, and the contextual changes that occur. This is even more so in the context of the impact of COVID-19, which has highlighted the question of just what European borders are, what they are like and what function, if any, they play.

**Author Contributions:** Conceptualization, J.V.R.; methodology, J.M.-U.; formal analysis, J.V.R.; writing—original draft preparation, J.M.-U. and J.V.R.; writing—review and editing, J.M.-U. and J.V.R. All authors have read and agreed to the published version of the manuscript.

**Funding:** This research was part of the project "Cross-border cooperation in Europe, geopolitics on a local scale. Analysis in five European countries of good practices for integration and global development (TRANSBORDEURCOP)" funded by Ministry of Economy of the Spanish Government, grant number 002424 (2016-2019).

**Institutional Review Board Statement:** Not applicable.

**Informed Consent Statement:** Not applicable.

**Data Availability Statement:** As a result of this project, data used is available in APTA (Territorial and Environmental Planning and Analysis) research group website, Department of Geography, Universitat de Girona, Spain: https://www.udg.edu/en/grupsrecerca/apta/recerca/resultats/projectes-interreg.

**Conflicts of Interest:** The authors declare no conflict of interest.

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
