# Peer review of "Territorial Development and Cross-Border Cooperation: A Review of the Consequences of European INTERREG Policies on the Spanish–French Border (2007–2020)"

_sustainability, doi:10.3390/su132112017_

Round 1
Reviewer 1 Report
This is an interesting and relevant paper which positions itself quite well into the academic and policy debates on cross-border cooperation. Its conclusions based on this case on the regional differentiation of the results of these CBC policies could however be strengthened and made more relevant for the wider topic of spatial inequalities within CBR’s when other research which focus on and further explain these spatial differences between the losing out of the people living close to the border and the population centres further away from the border would be mentioned. Similar observations on these kind of discrepancies were for instance made in studies on the PAMINA cross-border region south of Karlsruhe a little more than 10 years ago.
Author Response
Thanks for the comments.
The review consisted of expanding the reflection on the territorial reality in the development of the CBC. Also in agreement with Reviewer 2, the discussions have been highlighted, in addition to using the suggestion on the PAMINA region as an example. Also an expansion of the bibliography with more recent papers has been done.
Reviewer 2 Report
The abstract should very briefly represent the purpose, methodology used and results of your research and introduce a theoretical and practical contribution of the presented article. Please re-write the abstract reflecting this comment. The literature basis of the text should include more state of the art new positions. Authors should add the literature review section to the paper with a more in-deep analysis of previous researches.
There is no discussion section in the paper. Authors should add it with the discussion about the comparability of the research with the new world results. It's worth to add limitation of the papers in the conclusion.
Author Response
Thanks for the comments.
First, and following the suggestions, the summary has been modified to improve the overall exposition of the work structures.
Second, the bibliography section has been expanded, with more current authors and citations to enrich the theoretical framework and discussions.
In reference to the discussions, they have been included in the final section, to reinforce the conclusions. References to other European realities and quotations have been included.